# Automatic quantification of tumor-stroma ratio as a prognostic marker for pancreatic cancer

**Pierpaolo Vendittelli**[1]*, **John-Melle Bokhorst**[1], **Esther M. M. Smeets**[1], **Valentyna Kryklyva**[1], **Lodewijk A. A. Brosens**[1], **Caroline Verbeke**[2], **Geert Litjens**[1]

**1** Department of Pathology, Radboud University Medical Center, Nijmegen, The Netherlands, **2** Department of Pathology, Oslo University Hospital, Oslo, Norway

* pierpaolo.vendittelli@radboudumc.nl

## Abstract

### Purpose

This study aims to introduce an innovative multi-step pipeline for automatic tumor-stroma ratio (TSR) quantification as a potential prognostic marker for pancreatic cancer, addressing the limitations of existing staging systems and the lack of commonly used prognostic biomarkers.

### Methods

The proposed approach involves a deep-learning-based method for the automatic segmentation of tumor epithelial cells, tumor bulk, and stroma from whole-slide images (WSIs). Models were trained using five-fold cross-validation and evaluated on an independent external test set. TSR was computed based on the segmented components. Additionally, TSR's predictive value for six-month survival on the independent external dataset was assessed.

### Results

Median Dice (inter-quartile range (IQR)) of 0.751(0.15) and 0.726(0.25) for tumor epithelium segmentation on internal and external test sets, respectively. Median Dice of 0.76(0.11) and 0.863(0.17) for tumor bulk segmentation on internal and external test sets, respectively. TSR was evaluated as an independent prognostic marker, demonstrating a cross-validation AUC of 0.61±0.12 for predicting six-month survival on the external dataset.

### Conclusion

Our pipeline for automatic TSR quantification offers promising potential as a prognostic marker for pancreatic cancer. The results underscore the feasibility of computational biomarker discovery in enhancing patient outcome prediction, thus contributing to personalized patient management.

**Data Availability Statement:** Data from dataset A B and C cannot be shared publicly and without limitation because of the type of agreement signed,

but researchers who meet the criteria for access to confidential data could access the data upon request. All Data from dataset D are available from the National Cancer Institute database (accession number phs000178, link https://portal.gdc.cancer.gov/projects/TCGA-PAAD). Requests about data access can be sent to Valerie Dechering at: valerie.dechering@radboudumc.nl.

**Funding:** P.V. has received funding from the European Union's Horizon 2020 research and innovation programme under grant agreement no 101016851, project PANCAIM. The funders had no role in study design, data collection and analysis, decision to publish, or preparation of the manuscript.

**Competing interests:** The authors have declared that no competing interests exist.

## Introduction

Pancreatic ductal adenocarcinoma (PDAC), usually referred as pancreatic cancer is now the seventh leading cause of cancer-related deaths worldwide [1] and will soon become the second leading cause of cancer-related death in Western society [2]. Europe has the highest burden of pancreatic cancer in the world, with 150000 new cases in 2018 and 95000 deaths/year. With an average survival of 4.6 months, PDAC, which boasts a 5-year survival rate of less than 5% [3], emerges as the deadliest cancer globally, resulting in patients to lose up to 98% of their healthy life expectancy.

Histopathology is considered the diagnostic gold standard for PDAC diagnosis and characterisation. With the digitization of histopathology slides, pathologists can assess the presence of the disease via a computer screen. However, the prognostic power of current pathological assessment is very limited and correlates poorly with patient outcome [4] as there is an absence of proven prognostic biomarkers that can help patient stratification.

Currently, following diagnosis, patient management for PDAC is mainly based on the well-known TNM [4] staging system developed by the Union International Contre le Cancer (UICC), which is now in the 8th edition. This system stratifies patients by grading the tumor size (T), severity of the spread into regional lymph nodes (N) and the status of other distant metastatis (M), but despite being widely used, it is considered unreliable as patients with the same TNM stage often present different prognosis [5].

Given the limitations of TNM staging for prognostication in clinical practice, there is a clear need for identifying reliable biomarkers that better correlate tumor characteristics with patient outcome. Recent advances in applying machine learning in digitized pathology to extract prognostic features have opened the door for discovering quantitative morphological biomarkers to improve prognostic stratification in pancreatic cancer. AI-derived features could be predictive for pancreatic cancer as recently demonstrated by Nimgaonkar et al. [6], who specifically focused on extracting morphological features from whole-slide images (WSI) and correlating these with survival. Nimgaonkar et al. focussed on patients who were treated with gemcitabine after surgery, therefore it is unknown the predictive value for patients treated with other therapies.

PDAC is a complex disease often characterised by tumor heterogeneity [7] and dense stroma, which has been suggested to play a critical role in tumor development, progression and response to therapy.

The tumor-stroma ratio (TSR) is a widely studied prognostic factor, and it represent the relative amount of tumor cells and tumoral stroma. For a series of solid tumors such as breast [8], lung [9] and colorectal [10, 11] cancers, TSR has proved to be an independent prognostic factor. TSR is a straightforward measure assessed by microscopic inspection of H&E tissue sections, where a high stromal component is typically associated with poorer prognosis. Recently, researchers have examined the role of TSR in PDAC, but findings have been inconsistent. While Leppanen et al [12]. did not observe any correlation between TSR and overall survival, Li et al. [13] identified TSR as an independent prognostic factor. Typically, TSR is evaluated at low magnification with a single microscopic field of view that contains the tumor area where stromal tissue is most extensive and the tumor covers all four corners. However, this approach may not be suitable for PDAC due to its highly heterogeneous tumor microenvironment. Furthermore, the manual assessment of the TSR also suffers from significant inter-observer variability which can impact reliability. In recent work, Geessink et al. [10] have shown that machine-learning-based quantification of TSR in colorectal cancer allows for reproducible and reliable extraction of TSR while maintaining its prognostic power. In PDAC, Li et al. [13] have explored a semi-automatic approach, where pathologists annotated the tumor bulk area

**Table 1. Overview of the four different dataset used in this study.**

| Dataset | Source | Patients | Slides | Staining | Purpose |
|---|---|---|---|---|---|
| A | Radboudumc | 16 | 16 | HE—IHC | Epithelium segmentation |
| B | Multicentric | 162 | 162 | HE | Tumor epithelium segmentation |
| C | Radboudumc | 29 | 29 | HE | Internal validation |
| D | TCGA | 161 | 187 | HE | External validation—Survival analysis |

for each slide and an algorithm subsequently assessed TSR. Although their work showed promising results, but the manual identification of the tumor bulk still allows for potential sources of bias and inter-observer variability.

In this study, we propose a multi-step machine learning-based pipeline for both automatic PDAC segmentation and TSR ratio estimation. We made the code available at: https://github.com/DIAGNijmegen/automatic-tsr-quantification-for-pdac.

## Materials and methods

### Datasets

This study consisted of two main tasks: tumor segmentation through epithelium segmentation, and the automatic quantification of TSR.

Four independent datasets were considered in this study (Table 1). Specifically, we included two datasets from Radboud University Medical Center (RUMC), a publicly available dataset from The Cancer and Genome Atlas (TCGA), and a private multicenter dataset. All data were fully anonymized before access was granted and ethics committee waived the requirement for informed consent. Each individual dataset is described in detail on the following sections.

**Dataset A.** 16 patients who underwent pancreatic surgery at Radboudumc after the year 2000 were collected (study approved from the local ethical committee of Radboudumc, CMO-2016-3045). For each patient, a single Formalin-Fixed Paraffin-Embedded (FFPE) tissue block was chosen in consultation with an experienced pathologist, typically representing the largest tumor area. The tissue block was sectioned, stained with Hematoxylin and Eosin (H&E), and scanned using a 3DHistech Panoramic 1000 scanner. Next, the section was destained, restained with CK8/18, and scanned using the same scanner. Anti-CK18/8 is a cocktail of monoclonal antibodies targeting Ck8 and Ck18 specifically, typically used as a marker for epithelial cells. Both cytokeratins are targeted and revealed during the same IHC procedure. The resulting stain is therefore monochromatic. CK8/18 was used in conjuction with DAB and counterstained with haematoxylin, which result in the epihtelial cells being highlighted in brown and all others cells being stained blue. This was done according to our established protocol [14]. Finally, the H&E-stained slide was registered with the corresponding CK8/18 using an existing registration algorithm [15]. Images size were all (272128 x 294144) at spacing 0.25$\mu$m. Example of this dataset is reported in Fig 1.

**Dataset B.** 162 patients with PDAC and personal history of colorectal, breast, ovarian, endometrial, prostate, gastric cancer, and/or melanoma that were selected from the Dutch Nationwide Pathology Databank (PALGA) with the approval of their Privacy Commission and Scientific Council. The nationwide retrospective database-wide search (LZV2018-9) contained information up to and including February 2018 (nationwide coverage since 1991). Patient materials were collected from 25 collaborating laboratories across The Netherlands. This study was approved by the local ethical committee of Radboudumc (CMO-2017-3780). Images size were all (272128 x 294144) at spacing 0.25$\mu$m.

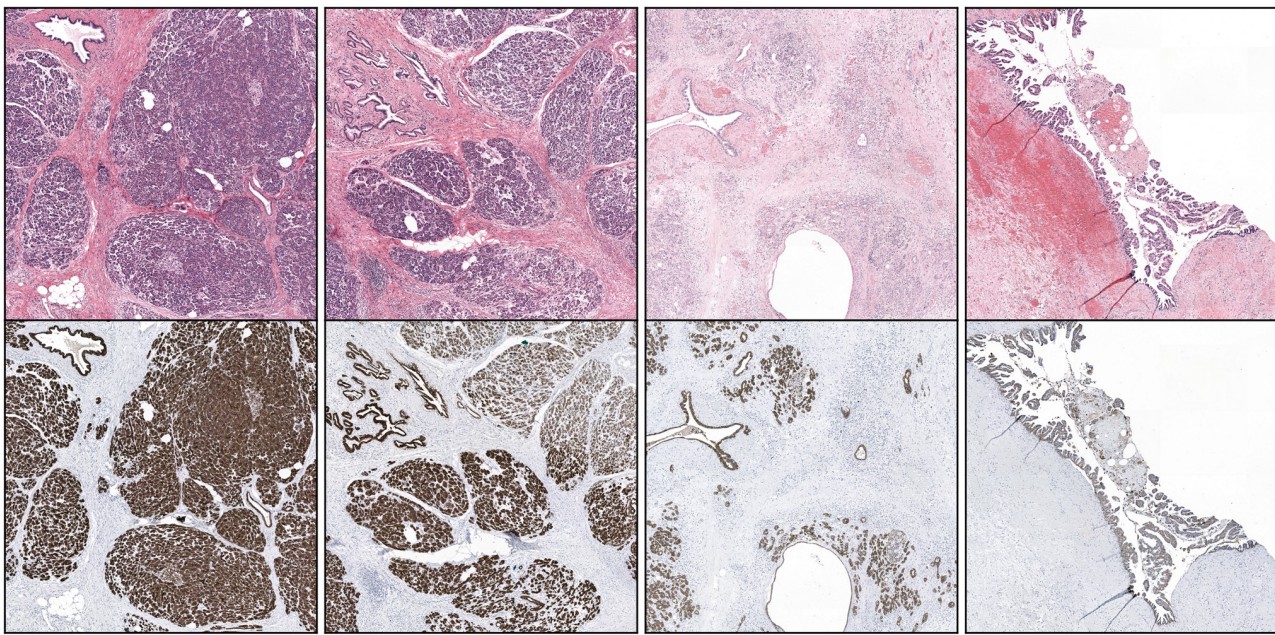

**Fig 1. Example of dataset A (first row H&E stained WSI, corresponding CK18/8 stained WSI on the second row).** The staining-restaining procedure results in perfectly matching slides.

**Dataset C.** We selected 29 patients from a PET-cohort of 158 patients with histologically proven PDAC who underwent [18F]FDG PET/CT in the period between 2004 and 2015 at Radboud University Medical Center, Nijmegen, The Netherlands during diagnostic work-up [16]. These patients were selected because whole tumor cross-sections were available for histological analysis. Images dimensions vary from (43008 x 80384) to (94720 x 96768) at spacing 1.0$\mu$m.

**Dataset D.** For external validation of the segmentation and survival analysis, we selected the public dataset from TCGA, study PAAD. This dataset is composed of 161 patients with 187 WSIs. An experienced pathologist made coarse tumor annotations on 35 randomly selected patients. In addition to coarse tumor annotations, 35 Regions of Interest (ROIs) of $2mm^2$ were annotated for tumor epithelium. Together with the slides, clinical and survival information for each patient were collected. We selected the following clinical variables: Age, Gender, Vital Status, Origin of the tumor, Primary diagnosis, Prior malignancy, and Survival time in days. Survival days was a variable represented either from $'days\text{-}to\text{-}death'$ if the patient died before the end of the study, or $'days\text{-}to\text{-}last\text{-}follow\text{-}up'$ if the patient was still alive at the end of the study. Images dimensions vary from (15347 x 15243) to (197207 x 84805) at spacing 0.25$\mu$m.

**Dataset preprocessing.** All the WSIs in the aforementioned dataset were preprocessed by first converting the available annotations into masks and then masking them with the tissue masks generated by a tissue-background segmentation algorithm [17] in order to remove the possible presence of the background. For the slides where annotations were not available, only tissue masks were generated to have faster both training and inference time.

## Methods

In this study, a two-step method for automatic tumor segmentation in WSIs of pancreatic cancer is proposed. A detailed pancreatic tumor segmentation is obtained by combining coarse

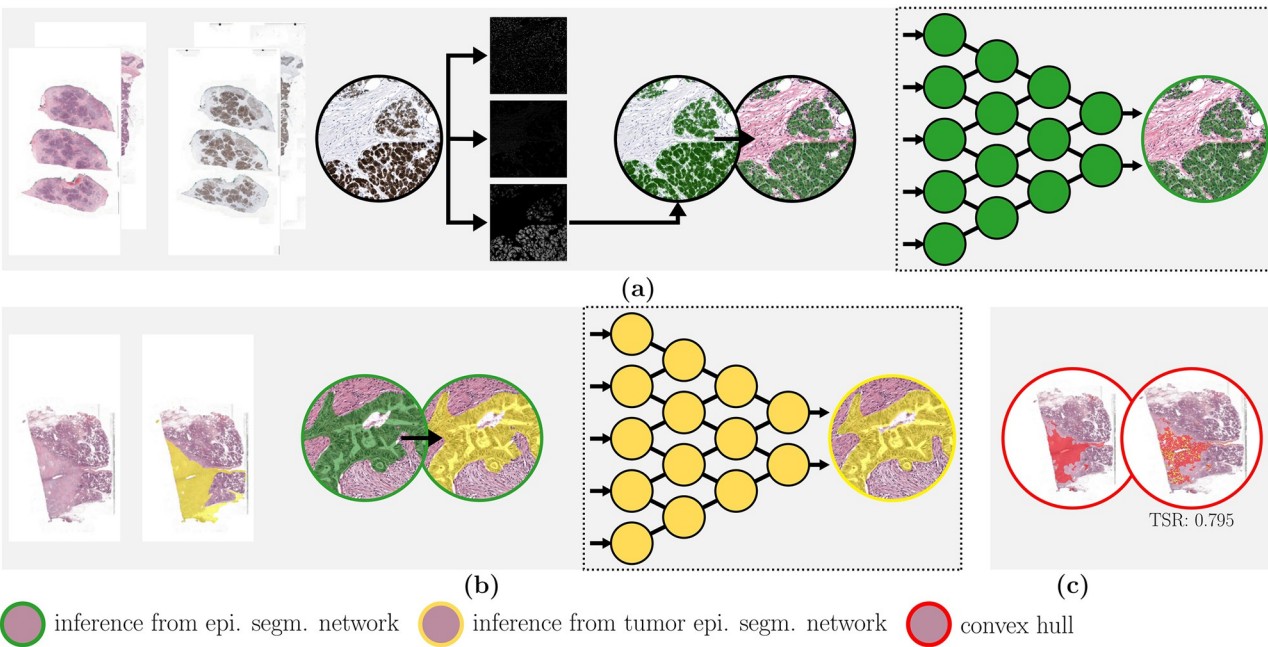

**Fig 2.** Flowchart highlighting different pipeline steps: (a) Epithelium segmentation and (b) tumor epithelium segmentation. Through the process of staining-destaining of paired H&E and IHC slides, epithelium annotations are obtained, which are then used to train an epithelium segmentation network (dataset A). This network annotates the rest of the slides. Subsequently, a tumor epithelium segmentation network is trained on the segmented epithelium combined with annotated tumor area (dataset B). Based on tumor epithelium segmentation, the tumor bulk is automatically determined by drawing a convex hull (c), on which TSR is calculated. Legend: In green the epithelium segmentation network, in yellow the tumor epithelium segmentation network and in red the resulting tumor bulk segmentation and TSR quantification.

tumor segmentation based on pathologists' annotations with immunohistochemistry-based epithelium segmentation, which requires developing several segmentation models. All models were constructed using the Pytorch version of the publicly available "Segmentation models" library [18]. An overview of the full pipeline is shown in Fig 2.

**Epithelium segmentation and convex hull.** The first step involved using dataset A to train an epithelium segmentation network. Tissue-background segmentation [17] was performed first, followed by the registration of the HE and IHC slides using a nonlinear image registration method [15]. Color deconvolution was applied to obtain an image only including the CK8/18 stain and subsequent thresholding converted it to a binary mask. The threshold value was empirically determined. A U-Net [19] model from the open-source Segmentation library [18] was used as the segmentation model. This network consisted of a Efficientnet-b0 encoder, pre-trained on ImageNet and a decoder with a depth of five. Skip connections were used between each encoder and decoder step. The network was trained to segment the epithelial cell in H&E using the registered binary mask resulting from the IHC slide as a reference.

In the second step, dataset B was used to train a tumor epithelium segmentation network. The ensemble of epithelium cell segmentation network obtained from the five-fold cross-validation over dataset A was applied to dataset B, and the results were masked with the tumor bulk annotations made by an expert pathologist. This resulted in precise tumor epithelium and normal epithelium annotations (Fig 2b). A U-Net based segmentation network with the same architecture as for epithelium segmentation was then trained to segment normal and tumor epithelium.

To evaluate the model's effectiveness in segmenting tumor epithelium, the Dice coefficient was calculated on the epithelium within the tumor areas, as presented in the experimental

setup section 1. In addition, the performance of tumor bulk segmentation was assessed by automatically generating a convex hull around the segmented tumor epithelium cells using the alphahull algorithm, with an empirically determined alpha value of 0.038. The Dice coefficient was computed between the pathologists' tumor bulk annotation and this convex hull.

**Automated tumor to stroma ratio quantification.** The overal goal of task 2 is to automatically quantify TSR and relate it TSR to patient survival. To calculate the TSR, segmentation of the tumor bulk, tumor epithelium and stromal components is needed. The former two were obtained through task 1, whereas the stromal segmentation was obtained by applying a multi-tissue segmentation algorithm, previously pre-trained on colorectal tissue [20], to the tumor bulk area.

Although TSR is not yet commonly assessed in pancreatic cancer, in colorectal cancer TSR is typically scored by 1) selecting a stroma-rich area within the tumor bulk at low resolution, 2) within that area identifying a field-of-view that is representative and surrounded by tumor cells, and subsequently 3) assessing the relative amount of stroma in 10% increments [21]. This process is complex and suffers from inter-observer variability. Despite those issues, in colorectal cancer this is a reliable procedure with prognostic power. However, in pancreatic cancer we have an additional challenge of high tumor heterogeneity. In this work, we circumvented this issue by not limiting the TSR to a single field-of-view, but by calculating it across the entire tumor bulk contained in one section. This follows a process similar to that of Li et al. [13], with the key difference that in our approach the tumor bulk was automatically segmented.

The automatic quantification of the TSR was carried out on dataset D, which included a set of clinical features that we used in combination with TSR to predict patient survival. The distribution of the automatic quantification of TSR in the slides analyzed shows a tendency towards very high values. The mean TSR is 0.731, and the median TSR is 0.852. A Logistic Regression model was trained with five-fold cross-validation to predict the probability of a patient's death within 6, 12, or 18 months and labels were assigned accordingly. In case multiple slides were available per patient, the slide with the highest TSR was selected.

**Experimental setup.** The epithelium segmentation network was trained using a five-fold cross-validation approach. Tiles sized 512x512 pixels were randomly selected from the WSIs in dataset A, with a resolution of 1.0 $\mu$m and a batch size of 10. For each experiment, training patches were augmented by applying various transformations, including horizontal and vertical flipping, blurring, random adjustments to HSV channels, contrast, and brightness.

To assess the performance, the Dice coefficient was computed at the WSI level. Specifically, for dataset A, encompassing the images used for training with available ground truth annotations, the Dice was calculated for the entire whole slide image. For dataset D, which involved 35 previously pathologist-annotated ROIs, the Dice was computed on these regions.

The tumor epithelium segmentation network underwent five-fold cross-validation, with training patches randomly taken from the WSIs in dataset B, at a 1.0 $\mu$m resolution and using a batch size of 10. Performance evaluation involved calculating the Dice coefficient on the whole slide image of dataset B. Additionally, for datasets C and D, one $2mm^2$ region within the tumor area, previously annotated by a pathologist, was used to calculate the Dice coefficient. Notably, for dataset D, both the epithelium segmentation network and the tumor epithelium segmentation network were assessed on the same region of interest.

Furthermore, the convex hull's performance was evaluated on datasets B, C, and D, utilizing the Dice coefficient. To enable a dependable comparison across these datasets, the parameter of the alphahull algorithm was kept constant.

## Results

### Tumor segmentation

**Epithelium segmentation.**   Table 2 presents the results of the epithelium segmentation task. The median Dice score for dataset A was 0.749 (0.30), while the median Dice score for dataset D was 0.717 (0.33). These results suggest that the algorithm performed similarly on both training and testing datasets. Fig 3 shows the performances of the epithelium segmentation model on dataset A. The network performs well at segmenting the epithelium, despite the fact of missing some glands. This is mostly due to the quality of the ground truth, which has been not refined to remove staining artefacts.

**Tumor epithelium segmentation.**   The results of the tumor epithelium segmentation task are presented in Table 2. The median Dice score for dataset B was 0.642 (0.25), while the median Dice scores for datasets C and D were 0.751 (0.15) and 0.726 (0.25), respectively. These results suggest that the algorithm performed similarly on all three datasets. The Dice coefficients were lower on the training set because the network tended to misclassify other types of epithelial cells, such as those from the duodenum, as tumor. This could be due to the fact that healthy pancreatic epithelium looks distinct from duodenum epithelium, which has a glandular composition that may be more similar to the cancerous pancreatic epithelium. Furthermore, as the evaluation on dataset B was conducted on the entire WSI, the presence of possible false positives was more visible, while on dataset C and D, this was less visible, being the algorithm evaluated on ROIs. Fig 4 provides qualitative examples of the Tumor segmentation network's performance.

**Tumor segmentation.**   The results of the Tumor segmentation task are presented in Table 2. The algorithm was tested on datasets B, C, and D. The median Dice score for dataset B was 0.7 (0.27), while the median Dice scores for datasets C and D were 0.76 (0.11) and 0.863 (0.17), respectively. These results suggest that the algorithm performed better on dataset D than on datasets B and C. A possible reason for this is that the on this dataset the tumor area in each slide covers on average more than 53% of the entire tissue, while on dataset B and C there is an average of approximately 40% of tumor to tissue area. Fig 5 shows an example on the convex hull generated on the segmented Tumor epithelium on dataset D.

### Survival analysis

The results of the survival analysis are presented in Fig 6. Based on the figure, TSR appears to be a promising prognostic factor for predicting 6-month survival in the patient population studied when combined with the aforementioned clinical variables. The area under the curve (AUC) of the cross-validation is 0.61±0.12, indicating moderate predictive performance. However, when predicting longer survival, the performance of the model decreased significantly, as

**Table 2. Results of the various tasks.**

| Task | Dataset | Median Dice (IQR) |
|---|---|---|
| Epithelium segmentation | A | 0.749 (0.30) |
| | D | 0.717 (0.33) |
| Tumor Epithelium segmentation | B | 0.642 (0.25) |
| | C | 0.751 (0.15) |
| | D | 0.726 (0.25) |
| Tumor segmentation | B | 0.700 (0.27) |
| | C | 0.760 (0.11) |
| | D | 0.863 (0.17) |

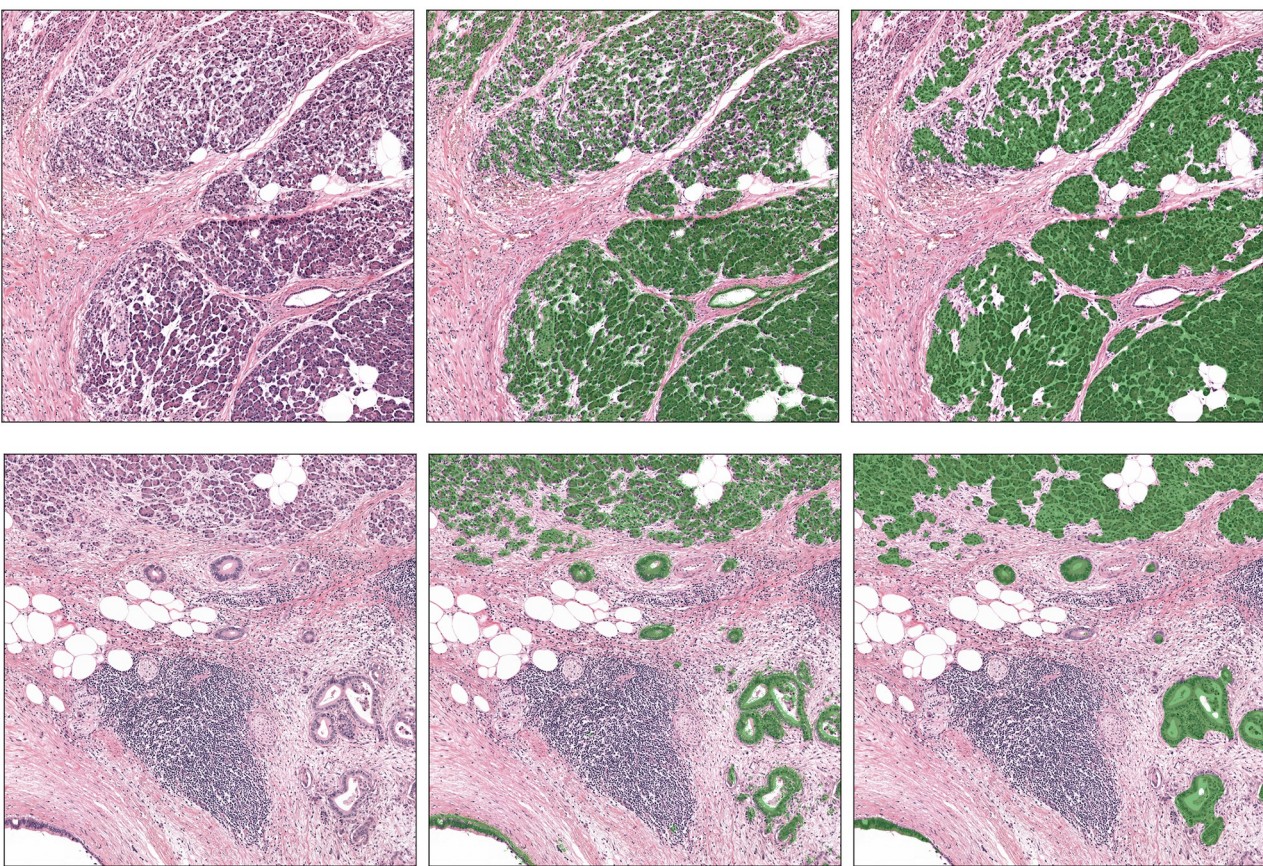

**Fig 3. Example of epithelial segmentation network on two different slides from dataset A.** Left column: example of two patches extracted from two different cases. Middle column: in green the overlay of the epithelium extracted from the CK18/8 stain. Right column: output of the network (in green the segmented epithelium). As we can see, lymphoid aggregates are correctly recognised as non-epithelium from the network.

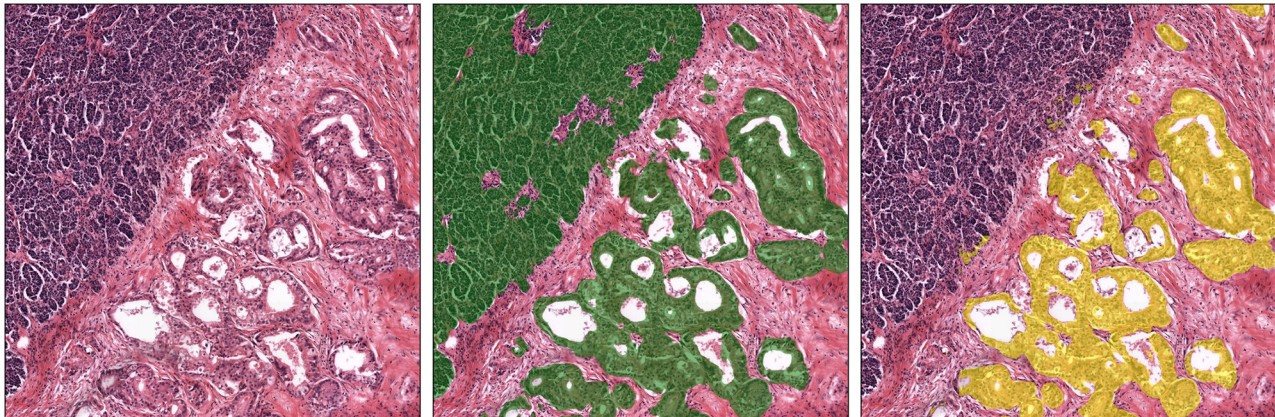

**Fig 4. Example of tumor epithelial segmentation network on a slide from dataset B.** First column: a randomly extracted patch from the dataset. Middle column: ground truth obtained by inferring the Epithelium segmentation network (top left cluster of epithelial cells are non cancerous, bottom right are cancerous). Right column: output of the tumor epithelium network. As we see, tumor epithelium is correctly segmented while healthy epithelium is discarded.

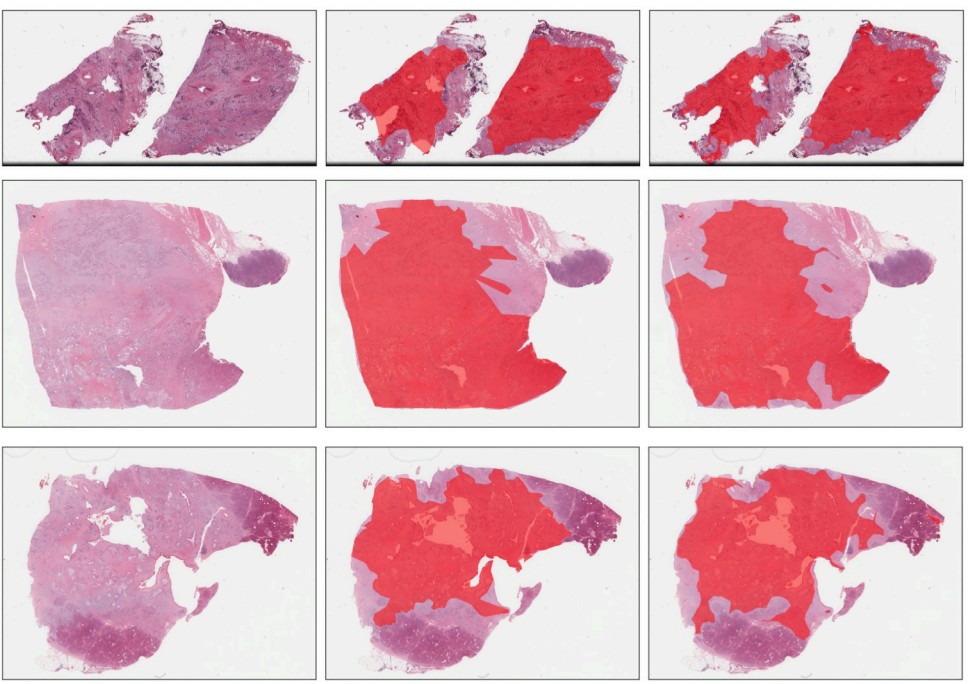

**Fig 5. Example of the performance of the alphahull algorithm on dataset D.** Middle column is the coarse tumor annotation made by pathologist, while right column is the convex hull automatically generated on the segmented tumor epithelium.

evidenced by the AUCs of 0.52±0.13 and 0.42±0.08 for 12-month and 18-month survival, respectively. Furthermore, we employed a Kaplan-Meier estimator, categorizing patients into high-risk and low-risk groups based on the TSR value, with a threshold set at 0.73—chosen as it represented the mean TSR value. The results are displayed in Fig 7. Although the Kaplan-

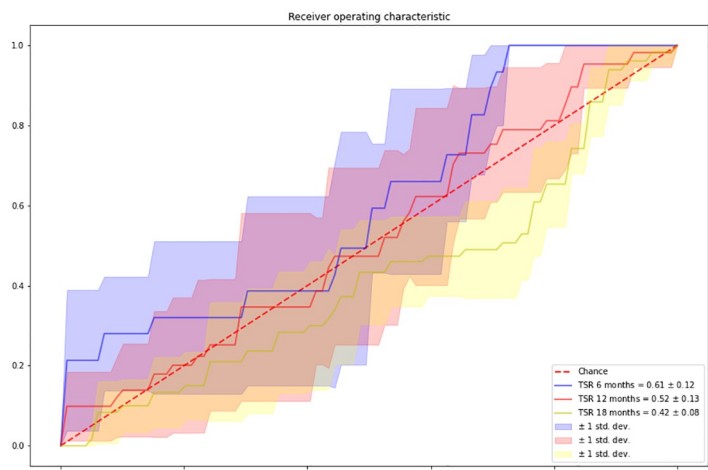

**Fig 6. AUC of the Logistic Regression model across five-fold cross-validation.** Shade in the area represents the standard deviation across all folds. The figure reports three experiments, 6 months, 12 months, and 18 months survival, respectively.

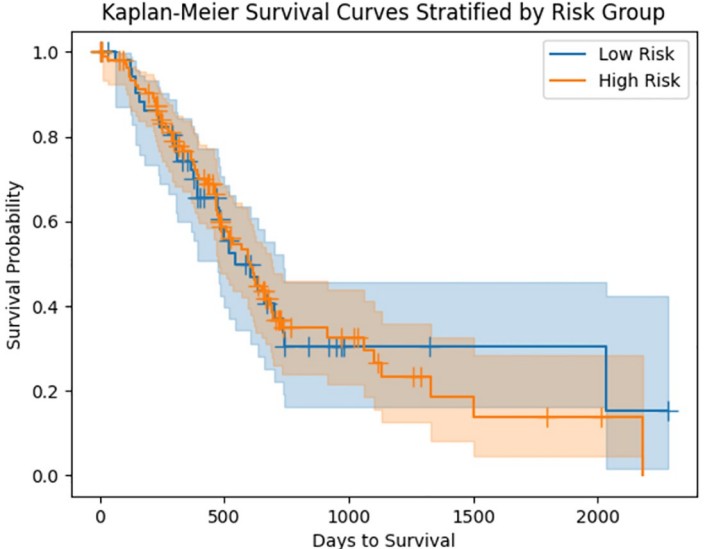

**Fig 7. Kaplan-Meier estimator.** Stratification was done using the mean TSR value.

Meier estimator indicates some differences in risks between patients, the significance level is not reached.

## Discussion and conclusion

We developed a deep learning multi-step method for the segmentation of tumor epithelial cells, tumor area, and TSR quantification in H&E stained slides of the pancreas. Common approaches for this task on H&E slides required manual epithelium annotations, which are time-consuming and prone to errors. We overcame this limitation by using a double-stained dataset, which allowed us to have nearly perfect epithelium ground truth.

Our algorithm showed reliable performance in segmenting epithelium, as demonstrated by cross-validation results and on independent validation over dataset D, with a mean Dice coefficient over the two datasets of 0.733 (0.3). With this reliable epithelium segmentation network, we automatically generated tumor epithelium annotations on dataset B, in which we had only coarse tumor annotations. Gao et al. [22] proposed a multi scale attention network to segment multiple tissues in pancreatic whole slide images and they reported a mean dice of 0.769 over five different types of tissues, including tumor epithelium, islets, ducts, blood vessels and nerves, while a mean dice of 0.757 on tumor epithelium, ducts and islets.

To further improve the segmentation of tumor epithelium, we trained a second network specifically suited for this task. We then used this network to automatically create a tumor area by drawing a convex hull around the segmented tumor epithelial cells. Results on different datasets (B,C,D) show that our network achieves good performance in segmenting tumor epithelium, with an average dice coefficient over the three datasets of 0.71 (0.22), with the performances of this network being better than the first network when evaluated on dataset D by 0.009. Finally, the Dice coefficients on the automatically generated convex hulls are reliable over all the tested datasets with an average Dice of 0.77 (0.18). Fu et al. [23] achieved similar results with an average Dice score of 0.803 on the patch-level segmentation task.

By generating a tumor area, we were able to automatically quantify TSR. We tested the prognostic power of TSR by combining it with other clinical variables and performing a

survival analysis at different time points. Our results suggest that TSR has potential predictive power for short-term survival (6 months) with an average AUC in the five-fold cross-validation of 0.61±0.12 (clinical variable alone 0.60±0.12), although this power decreases as the time slot interval increases. We further investigated the possible stratification between high-risk and low-risk patients, and our results lean towards the assumption that higher TSR is associated with higher risk, while lower TSR is associated with lower risk. However, we should note that the survival analysis did not provide statistical significance. Chen et al. [24] conducted a similar analysis on the same dataset, achieving an AUC of 0.58 for predicting survival on WSI only and 0.65 when combining the WSI with genomic information.

Despite the promising results, our work suffers from several limitations. For instance, the double-stained dataset is relatively small, which prevented us from having a proper test set and required instead cross-validation. Additionally, the color deconvolution has not been corrected for staining artefacts, which could affect the performance of our algorithm. The tumor epithelium network suffers from false-positive over-segmentation, meaning that our algorithm could segment other tissue epithelium (e.g. healthy duodenum) as tumor. Finally, despite the potential prognostic importance of TSR in predicting short-term survival, our experiments did not achieve statistical significance. It is important to note that the predictive performances of TSR seem to decrease substantially when predicting longer survival. Further analysis with larger cohorts of patients is required to fully validate the prognostic value of TSR when combined with clinical variables.

By eliminating the need for manual assessment, our approach can increase efficiency and reduce the potential for human error. Furthermore, the ability to quantify TSR automatically over the entire tumor area provides a more comprehensive assessment of this potential biomarker.

Overall, the results of our study demonstrate the potential of automated approaches for the assessment of TSR as a prognostic biomarker. While further validation is required, the results presented here provide a foundation for future research and development in this area. In future work, it would be interesting to explore additional biomarkers, such as the presence of tumour-infiltrating lymphocytes, in combination with TSR.

## Acknowledgments

Authors would like to thank Stan Noordman for its contribution in the creation of Fig 2.

## Author Contributions

**Conceptualization:** Pierpaolo Vendittelli, Geert Litjens.

**Data curation:** Esther M. M. Smeets, Valentyna Kryklyva, Lodewijk A. A. Brosens, Caroline Verbeke.

**Funding acquisition:** Pierpaolo Vendittelli.

**Methodology:** Pierpaolo Vendittelli, Geert Litjens.

**Resources:** Pierpaolo Vendittelli, Geert Litjens.

**Software:** Pierpaolo Vendittelli, John-Melle Bokhorst.

**Supervision:** Geert Litjens.

**Validation:** Pierpaolo Vendittelli, Caroline Verbeke.

**Visualization:** Pierpaolo Vendittelli.

**Writing – original draft:** Pierpaolo Vendittelli.

**Writing – review & editing:** John-Melle Bokhorst, Esther M. M. Smeets, Valentyna Kryklyva, Lodewijk A. A. Brosens, Caroline Verbeke, Geert Litjens.

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
