## [Decision Letter · Decision Letter 0]

8 Jan 2024

PONE-D-23-29250Automatic quantification of TSR as a prognostic marker for pancreatic cancer.PLOS ONE

Dear Dr. Vendittelli,

Thank you for submitting your manuscript to PLOS ONE. After careful consideration, we feel that it has merit but does not fully meet PLOS ONE’s publication criteria as it currently stands. Therefore, we invite you to submit a revised version of the manuscript that addresses the points raised during the review process.

We look forward to receiving your revised manuscript.

Kind regards,

Shuai Ren

Academic Editor

PLOS ONE

Journal Requirements:

Reviewers' comments:

Reviewer's Responses to Questions

**Comments to the Author**

1. Is the manuscript technically sound, and do the data support the conclusions?

Reviewer #1: Yes

2. Has the statistical analysis been performed appropriately and rigorously? 

Reviewer #1: No

3. Have the authors made all data underlying the findings in their manuscript fully available?

Reviewer #1: No

4. Is the manuscript presented in an intelligible fashion and written in standard English?

Reviewer #1: Yes

5. Review Comments to the Author

Reviewer #1: It’s a relatively straightforward manuscript to follow: the study focuses on automatic quantifications of tumor-stroma ratio (TSR) based on U-net neural network architecture for Pancreatic ductal adenocarcinoma (PDAC) and attempts to examine reproducibility across multiple cohorts. Clinically, it attempts to examine the prognosis association of TSR for PDAC. The overall technical reproducibility across multiple datasets appears to be very robust, while the survival association is marginal.

While the technology seems to be mainly applying existing U-net, the major contribution is probably using H&E stain-destain-restain IHC to guide learning, for which I have to applaud as a practical value as time and cost for human experts to curate these annotations. However, the authors may want to spell out the exact biology specificity of CK18/8 (is that co-stain of two markers) for PDAC, as not all readers are experts in the disease. Also, how can such CK18/8 guidance help to tease out normal versus tumor epithelial cells (as partially shown in Fig 4). Also, can this protocol be expanded to accommodate co-stain of immune cell markers? I speculate that TSR together with immune scoring (e.g. TIL- tumor infiltrating lymphocytes) would be more prognosis-relevant, especially considering the ambiguities that a equal high proportion of TSR may mean immune-“hot” or immune-“cold” tumor microenvironment.

As major concern: No codebase and no trained model released on GitHub? I understand the imaging data can be sensitive to share, but I do expect some codebase and trained model availability as a computational paper. illustration using the publicly available TCGA PDAC dataset. I hope it’s reasonable for anyone who wants to reproduce or reuse this work.

Maybe some statistical question: why not use a censored survival association model and report C-index instead of logistical regression?

A few minor issues: 1. It’s better to spell out the full name of TSR in the title of the manuscript. 2. I don’t quite understand the eligibility of dataset-B, why it requires both PDAC and history of other cancers? “162 patients with PDAC and personal history of colorectal, breast, ovarian, endometrial, prostate, gastric cancer,”

6. PLOS authors have the option to publish the peer review history of their article (what does this mean?). If published, this will include your full peer review and any attached files.

Reviewer #1: No

---

## [Author Response · Author response to Decision Letter 0]

26 Feb 2024

Reviewer's Comment #1: The authors may want to spell out the exact biology specificity of CK18/8 (is that co-stain of two markers) for PDAC, as not all readers are experts in the disease.

Response: “Anti-CK18/8 is a cocktail of monoclonal antibodies targeting Ck8 and Ck18 specifically, typically used as a marker for epithelial cells. Both cytokeratins are targeted and revealed during the same IHC procedure. The resulting stain is therefore monochromatic.” We added this information in Subsection Datasets (dataset A) of the paper. 

Reviewer's Comment #2: Also, how can such CK18/8 guidance help to tease out normal versus tumor epithelial cells (as partially shown in Fig 4).

Response: The IHC stain is used to generate accurate labels in order to train a first Epithelium segmentation network. The differentiation between tumor and healthy epithelial cells is done through application of this epithelium segmentation network over dataset B, for which tumor bulk area was coarsely annotated under the supervision of an expert pathologist, under the assumption that all the epithelial cells inside the annotated tumor area are classified as cancerous.

Reviewer's Comment #3: Can this protocol be expanded to accommodate co-stain of immune cell markers? I speculate that TSR together with immune scoring (e.g. TIL- tumor infiltrating lymphocytes) would be more prognosis-relevant, especially considering the ambiguities that a equal high proportion of TSR may mean immune-“hot” or immune-“cold” tumor microenvironment.

Response: A combination of CK18/8 and, for example, a CD3 IHC for lymphocytes would certainly be possible. A possible approach would see the extension of the first Unet to segment the epithelium and the TILs, while the second Unet would segment the tumor epithelium, the healthy epithelium and the TILs. Survival analysis would be done using the combination of the two biomarkers. This is an interesting approach and would certainly be considered in future work, but given the significant extra work and cost (recutting tissue blocks, restaining, and developing new models), we did not include these experiments in this work. We did include a few sentences in the discussion:

“In future work, it would be interesting to explore additional biomarkers, such as the presence of tumour-infiltrating lymphocytes, in combination with TSR.”

Reviewer’s Comment #4: No codebase and no trained model released on GitHub? I understand the imaging data can be sensitive to share, but I do expect some codebase and trained model availability as a computational paper. illustration using the publicly available TCGA PDAC dataset. I hope it’s reasonable for anyone who wants to reproduce or reuse this work.

Response: We thank the reviewer for this comment, and we have addressed this limitation by making available our Docker for public usage. It can be found at the following link: https://github.com/DIAGNijmegen/automatic-tsr-quantification-for-pdac

Reviewer’s Comment #5: Maybe some statistical question: why not use a censored survival association model and report C-index instead of logistical regression?

Response: We expanded the survival analysis by adding the Kaplan-Meier estimator. Results of this have been added in the paper, section Results, subsection survival analysis. 

Figure 7 Kaplan-Meier estimator. Stratification was done using the mean TSR value.

We added the following sentences in the paper:

“Furthermore, we employed a Kaplan-Meier estimator, categorizing patients into high-risk and low-risk groups based on the TSR value, with a threshold set at 0.73—chosen as it represented the mean TSR value. The results are displayed in Fig 7. Although the Kaplan-Meier estimator indicates some differences in risks between patients, the significance level is not reached.”

Reviewer’s Comment #8: It’s better to spell out the full name of TSR in the title of the manuscript.

Response: We have addressed the comment from the reviewer changing the name of the manuscript, including tumor-stroma ratio instead of TSR. 

Reviewer’s Comment #9: I don’t quite understand the eligibility of dataset-B, why it requires both PDAC and history of other cancers?

Response: Dataset B comes from an existing research study into pancreatic cancer by researcher V.K. In that study they used the availability of the history of previous cancer as an inclusion criterion, which we mention for completeness. In our study we did not use this information in any way.

Editor’s Comment a: If there are ethical or legal restrictions on sharing a de-identified data set, please explain them in detail (e.g., data contain potentially sensitive information, data are owned by a third-party organization, etc.) and who has imposed them (e.g., an ethics committee). Please also provide contact information for a data access committee, ethics committee, or other institutional body to which data requests may be sent.

Response: The data from datasets A, B, and C cannot be publicly shared without limitation due to the terms of the agreement in place, and the presence of sensible information in data. However, researchers who meet the criteria for accessing confidential data may request access by submitting a detailed request outlining the scope of their research. This request will be forwarded to the ethical committee for approval. As outlined in the paper, the study received approval from the local ethical committee of Radboudumc (CMO-2016-3045, CMO-2017-3780).

Editor’s Comment b: If there are no restrictions, please upload the minimal anonymized data set necessary to replicate your study findings as either Supporting Information files or to a stable, public repository and provide us with the relevant URLs, DOIs, or accession numbers.

Response: All Data from dataset D are available from the National Cancer Institute database (accession number phs000178, link https://portal.gdc.cancer.gov/projects/TCGA-PAAD.).

Dataset D was used to perform survival analysis, and researchers who want to replicate results can download the data from the public repository and apply the algorithm published on github as specified on response to Reviewer’s Comment #4.

---

## [Editor Report · Decision Letter 1]

27 Mar 2024

Automatic quantification of tumor-stroma ratio as a prognostic marker for pancreatic cancer.

PONE-D-23-29250R1

Dear Dr. Vendittelli,

We’re pleased to inform you that your manuscript has been judged scientifically suitable for publication and will be formally accepted for publication once it meets all outstanding technical requirements.

Kind regards,

Shuai Ren

Academic Editor

PLOS ONE
---

## [Editor Report · Acceptance letter]

10 May 2024

PONE-D-23-29250R1 

PLOS ONE

Dear Dr. Vendittelli, 

I'm pleased to inform you that your manuscript has been deemed suitable for publication in PLOS ONE. Congratulations! Your manuscript is now being handed over to our production team.

Kind regards, 

on behalf of

Dr. Shuai Ren 

Academic Editor

PLOS ONE